# Perceptions, attitudes, awareness and perspectives towards sustainability practices and climate change among nurses: a systematic review protocol

Ebenezer Akore Yeboah [ORCID],[1] Amanda Rodrigues Amorim Adegboye,[2] Rosie Kneafsey[1]

¹Research Centre for Healthcare and Communities, Coventry University, Coventry, UK
²Centre for Agroecology, Water and Resilience, Coventry University, Coventry, UK

**Correspondence to**
Ebenezer Akore Yeboah;
ebenezer.akore@gmail.com

## ABSTRACT

**Introduction** Climate change has been described as the most significant threat to humanity and human health to have emerged this century. It is widely accepted that contemporary human activities are the major causes of climate change. It is also acknowledged that damaging human activities could be amenable to change through proactive environmental behaviours. Healthcare professionals have the potential to promote climate advocacy and mitigation through collective effort and individual actions. However, research suggests that nurses may not be aware of their potential to effect positive action. This review will synthesise evidence regarding nurses' perceptions, attitudes, awareness and perspectives towards sustainable nursing practices and climate change.

**Methods and analysis** The Joanna Briggs Institute (JBI) methodology for mixed-methods systematic reviews will be applied to this proposed systematic review. It will be reported in accordance with the Preferred Reporting Items for Systematic Reviews and Meta-Analyses guidelines. CINAHL, PsycINFO, SCOPUS and PubMed databases will be searched. Data appraisal will be completed using the JBI and Mixed Methods Assessment Tool critical appraisal tool. Data synthesis and integration will follow the JBI convergent integrated approach.

**Ethics and dissemination** In compliance with university ethics requirements for secondary research and postgraduate researchers, ethical approval will be sought from the Coventry University Ethics Committee, UK. Dissemination of findings will be achieved through peer-review publications, conference presentations and seminars with local, national and international audiences.

## STRENGTHS AND LIMITATIONS OF THIS STUDY

⇒ This study will be limited to papers published in English.
⇒ There is a risk that relevant information in other languages could be missed.
⇒ Rigour and transparency will be observed in the review process.

## INTRODUCTION

Climate change has been described as the most significant threat this century to humanity and the greatest source of harm to human health.[1] Climate change is defined as 'the systematic change in the long-term state of the atmosphere over multiple decades or longer'.[2] In the 19th century, natural events such as changes in incoming solar radiation, volcanoes and changes in global biogeochemical cycles were identified by scientists as the major causes of climate change.[3] However, scientific consensus has established that contemporary human activities such as combustion of fossil fuel for energy production and energy use, deforestations, atmospheric aerosols, change in land use and emission of other gases such as water vapour and other greenhouse gases are the major causes of climate change.[3]

The impact of climate change on human health is significant and as such is a critical issue.[4] It is widely accepted that the populations which suffer most are those which contribute little to greenhouse gas emissions.[5] Such populations are particularly vulnerable due to geographical placement, social isolation and economic disadvantage. The documented impact of heatwaves, floods, drought and climate change-related disease have been devastating. For example, it is estimated that climate change will cause an additional 250 000 deaths across the globe between 2030 and 2050, from heat exhaustion in elderly people, malnutrition among children, malaria and diarrhoea.[4]

Globally, health professionals are acknowledging and responding to the damage caused by climate change and the impact on health.[6] At the same time, healthcare systems themselves account for 4.4% of the global net $CO_2$ emissions thus rendering significant climate impact, both through carbon emissions and

use of carbon compounds for human health benefit. Indeed, it has been identified that if the healthcare sector was a country, it would rank the fifth-largest producer of greenhouse gases on the planet.[7] More specifically for the UK, the National Health Service accounts for 5.4% of the UK greenhouse emissions and this is higher than the global average of the healthcare system.[7]

Healthcare professionals have the potential to promote climate advocacy and mitigation through collective effort and individual actions.[8] However, research suggests that nurses may not be aware of the potential to effect positive action.[9] A study in Sweden identified that while nurses exhibited an understanding of environmental issues in their local setting, a comprehensive global perspective on climate change and environmental issues was less developed.[10] In contrast, a study in China showed that while 76% of nurses knew that climate change would affect public health, more than 50% did not know that their own local nursing practice could be related to carbon compound emissions and subsequently climate change.[11] Across continents, a study including American nurses found that they clearly believed the present threat of climate change and recognised their responsibility as health workers to address the health effects of climate change.[9]

It is known that nurses form the largest proportion of the healthcare workforce and are a significant professional force. As such, practising environmentally responsible healthcare as well as embedding the concepts of sustainability and climate actions into nursing curricula could be important in the fight against climate change.[12] There exists an ongoing integrative review protocol exploring awareness and attitudes of student nurses and educators toward sustainability and climate change.[13] However, there is no systematic review focusing on the perspectives of registered/qualified nurses. Studies have used different samples, research designs and have reached different conclusions. Therefore, this review aims to systematically synthesise evidence concerning nurses' perceptions, attitudes, awareness and perspectives of sustainable nursing practice and climate change.

## METHOD

### DESIGN
A mixed-method systematic review will be conducted to answer the research question related to the phenomena of interest in this study. Mixed-method systematic review merges and homogenise findings of quantitative and qualitative studies within a single review.[14] Mixed-methods reviews are useful in producing systematic reviews of direct importance or significance to policy-makers and practitioners.[15] This design is also appropriate and suitable when the researcher wants to broaden the conceptualisation of evidence that is methodologically inclusive. It synthesises evidence that will be available to a wider range

of consumers. Hence, this approach enables a comprehensiveness exploration of the literature related to nursing, climate change and sustainability.[16] The updated JBI methodology for mixed-methods systematic reviews[17] will be used for this systematic review. It will be reported in accordance with the Preferred Reporting Items for Systematic Reviews and Meta-Analyses (PRISMA) guidelines.[18] The review protocol is registered in the Open Science Framework and registration DOI 10.17605/OSF. IO/8H3TC.

### Identification of problem
The Sample, Phenomenon of Interest, Design, Evaluation and Research type framework has been used to formulate the research question and determine the criteria for inclusion and exclusion (table 1).

All studies regardless of the date of publication will be included. Studies across the globe or any country will be included but limited to publications in English.

### Searching literature
#### Search strategy
A preliminary scoping search was done to identify articles on the study topic. A full search strategy was developed after filtering through various titles and abstract of relevant articles and their index terms (table 2, see also online supplemental file).

#### Information source
The databases to be searched include; CINAHL, PsycINFO, SCOPUS and PubMed. These databases have been selected due to their broad range of literature in the fields of health and social sciences, nursing and climate change.

#### Study selection
The search results from each database will be imported in EndNote reference manager to remove duplicates and then transferred to Rayyan web tool where screening based on the inclusion and exclusion criteria will take place. In the initial phase, 20% of the studies will be screened by the three reviewers (EAY, ARAA and RK) independently based on title and abstract. This independent screening process will reduce the risk of bias and ensure consistency in the review process.[19] Once the initial screening phase is done, two reviewers (EAY and ARAA) will continue the title and abstract screening on the remaining studies. Any studies selected at this stage will then be subjected to full text screening by the same two reviewers (EAY and ARAA).[13] Any differences between the two reviewers will be resolved through discussion or by the third reviewer (RK).

During this second search, bibliography or reference lists of the studies which were included will be manually reviewed. Any reason for excluding a paper during the full text assessment that does not meet the inclusion criteria will be recorded and reported in the systematic review findings. In instances of disagreement between the two reviewers in the second phase of screening the full text,

**Table 1** Inclusion and exclusion criteria in the literature search

| Domains | Inclusion | Exclusion |
|---|---|---|
| Sample | ▶ Studies/publication/articles focused on registered or qualified nurses regardless of work setting<br>▶ Can include other health professionals as long as findings reported nurses separately. | ▶ Studies/publication/articles on climate change focusing on nursing students and/or nursing educators.<br>▶ Studies on other healthcare professional for example, care assistant, doctor, etc. |
| Phenomenon of interest | ▶ Climate change, environmental sustainability, net-zero healthcare and nursing practice or any related terms to these | ▶ Studies that do not include climate change or sustainable environmental practice or any related term |
| Design | ▶ Empirical research studies using any research design, for example, interview, survey, focus group, etc.<br>▶ Published literature, governmental reports, technical documents, conference abstracts, manuals, on net-zero healthcare (or climate change) with relation to nursing (which are primary or empirical study). | ▶ Commentaries, press releases, speeches, letters to the editor are excluded<br>▶ All publications relating to the phenomena of interest which are not empirical research. |
| Evaluation | ▶ Studies exploring the perception, OR attitude, OR awareness (or knowledge) OR the perspective on climate change and related terms | |
| Research type | ▶ Qualitative, quantitative or mixed-methods approaches to primary research | ▶ Secondary research |

consensus will be reached through discussion or third reviewer. All members of the review team (EAY, ARAA and RK) will inspect and assess the final list of papers that met the inclusion criteria before the progressing to the next stages of the review methodology. A PRISMA flow diagram will be designed to show the completed search.[20]

### Assessment of methodological quality

The standardised JBI qualitative critical appraisal tool will be used to assess qualitative papers that meet the inclusion

criteria.[21] The quantitative and mixed-method papers will be methodologically appraised with the Mixed Methods Assessment Tool version 2018.[22] These methodological appraisals will be done by the two review members (EAY and ARAA) independently, prior to data extraction.

The quality appraisal of the included papers will be compared and discussed to determine the overall quality score and in situations of dissimilarity, it would be discussed collaboratively among the review team,

**Table 2** Search terms

| | Concept | Truncated terms |
|---|---|---|
| Sample | "nursing" OR "nurses" | "nurs*" |
| Phenomenon of interest | "climate change" OR "sustainability" OR "sustainable healthcare" OR "environmentally responsible healthcare" OR "global warming" OR "environmental responsibility" OR "net zero healthcare" | "climate change" OR "sustainab*" OR "sustainable healthcare" OR "environmentally responsible healthcare" OR "global warming" OR "environmental responsib*" OR "net zero healthcare" |
| Design | "Survey" OR " questionnaire" OR "interviews" OR "observational study" OR "case study" OR "focus group discussion" | "Survey*" OR " questionnaire*" OR "interview*" OR "observ*" OR "case stud*" OR "focus group*" |
| Evaluation | "perception" OR "awareness" OR "perspectives" OR "attitude" OR "beliefs" OR "knowledge" OR "opinion" OR "sustainable practice" OR "practice" | "Opinion*" OR "perce*" OR "aware*" OR "perspect*" OR "attitude*" OR "belie*" OR "sustainable pract*" OR "know*" OR "pract*" |
| Research type | "mixed methods" OR "qualitative" OR "quantitative" | "mixed method*" OR "qualitative" OR "quantitative" |

employing triangulation or an independent reviewer as a measure of credibility or trustworthiness.[19] This will be done to reduce any risks of bias and provide compelling evidence for the study. Regardless of the methodological quality, all included studies will undergo data extraction and synthesis (where possible). The results of the appraisal will summarised in a table and reported in the review.

### Data extraction

Both the qualitative and quantitative data will be extracted from the included papers by two reviewers (EAY and ARAA) using the standardised JBI data extraction tools.[21] Any discrepancies will be resolved through discussion among the two reviewers or with triangulation. Authors of papers will be contacted for missing or additional data, if required.[21]

### Data transformation

For the purpose of integration, the extracted data will need to be transformed into mutually compatible forms.[23] In this review, the retrieved quantitative data will undergo qualitising.[24] In this way, extracted quantitative data will be converted into qualitative data. This will involve conversion of quantitative data into textual description thus themes, categories, etc, in order to be able to answer the review question. Qualitising is chosen to reduce the risk of error as opposed to quantitising.[24]

### Data synthesis and integration

The JBI convergent integrated approach will be adopted to synthesise the transformed data.[21] This involves combining the extracted data from quantitative and qualitative or mixed-method studies. The qualitised data will be assembled with qualitative information and then categorised. The categorised data will be pooled together based on similarity in meaning to produce a set of integrated finding.[17]

### Patient and public involvement

None.

## DISCUSSION

This mixed-method review will enrich the conceptualisation of evidence so that methodologically inclusive and synthesised data will be available to the nursing community and policy-makers. This review will synthesise the existing body of literature related to nurses' perceptions, attitudes, awareness and perspectives towards sustainability practices in relation to climate change. The findings could be significant to hospital managers, nurse leaders and green or sustainability champions when planning environmentally responsible healthcare projects. For example, relating to sustainable healthcare practices, meeting net-zero goals or greening hospitals. Since nurses form the largest part of the healthcare workforce and present a significant professional force, translating evidence from research into practice and

promoting nurse engagement in this aspect of healthcare could be impactful.[12 25] The review will be disseminated in journals, conferences and with relevant institutions to improve practice and share knowledge for the benefit of the global population. Climate change and sustainability are becoming ubiquitous in all spheres of life as the world experiences the impact of this phenomenon.

## CONCLUSION

The findings of this mixed-method review will be to raise awareness of sustainable healthcare actions and identify opportunities to accelerate the implementation of environmentally responsible nursing practices. The review will also promote dialogue with policy-makers, nursing and healthcare leaders regarding the benefit of further mobilising nurse engagement in climate actions and policies.

### Ethics

Ethical approval will be sought from the University Ethics Committee, Coventry University, UK.

**Acknowledgements** Thank you to all the staff of Research Centre for Healthcare and Communities, Coventry University, for supporting the PhD study.

**Contributors** EAY, ARAA and RK, all contributed to the conception, design, drafting manuscript, revising and approval of the final version to be published. However, EAY is accountable for all aspect of the work.

**Funding** Thank you to the Research Centre for Healthcare and Communities, Coventry University, for funding the PhD study. No fund/grant number.

**Competing interests** None declared.

**Patient and public involvement** Patients and/or the public were not involved in the design, or conduct, or reporting, or dissemination plans of this research.

**Patient consent for publication** Not applicable.

**Provenance and peer review** Not commissioned; externally peer reviewed.

**ORCID iD**
Ebenezer Akore Yeboah http://orcid.org/0000-0003-4365-7750

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
