## [Reviewer comments · BMJ Open]

ARTICLE DETAILS

TITLE (PROVISIONAL)	Perceptions, attitudes, awareness and perspectives towards sustainability practices and climate change among nurses: A systematic review protocol
AUTHORS	Akore Yeboah, Ebenezer; Adegboye, Amanda; Kneafsey, Rosie

VERSION 1 – REVIEW

REVIEWER	Magin, Parker University of Newcastle, School of Medicine and Public Health
REVIEW RETURNED	24-Feb-2023

GENERAL COMMENTS	This protocol reports the methodology for a systematic review of an important topic - nurses and the nexus of their clinical practice and environmental sustainability. The rationale for the importance of the topic is well presented. The protocol mainly follows a conventional methodology, but there are a few points to be addressed. The inclusion of conference abstracts and reports (not closely defined) is potentially problematic – especially with inclusion of studies in the review regardless of data being able to be extracted or of assessed study quality. The manuscript needs careful review of English usage and syntax. Clumsy language and grammar is frequent. For example, 'But these nurses did not have the aptness to these climate change health implication...' 'As nurses form the largest part of the health professional group, it is urging for nurses to find mitigating factors...' '...it seems relevant to systematically synthesis the literature and critical appraise the studies in this field.' I'm not sure why the team will be seeking ethics approval for a systematic review.
--

REVIEWER	Naughton, Bernard Trinity College Dublin, Pharmacy and Pharmaceutical Sciences
REVIEW RETURNED	08-Mar-2023

GENERAL COMMENTS	Thank you for submitting your protocol for review. Please find my suggestions for improvement below. 1. The PRISMA-P checklist does not contain page or line numbers to direct the reader to where in the article the PRISMA-P criteria is located, I suggest these are added.2. Also the PRISMA-P checklist is presented on a Biomed Central form. I suggest you submit a standard PRISMA-P statement rather than one customized from a different journal.
---

	3. I notice that you will be applying for ethical approvals for your study. Usually, an ethics application is not required for a systematic review as you will be dealing with secondary data. I suggest you clarify this point with your institution. 4. You mentioned that you will be taking a mixed method systematic review approach and will analyse both qualitative and quantitative data. However, in your PRISMA-P statement under point 15a you mention that quantitative data won't be synthesized. Please could you clarify this? Finally, I would like to wish you all the very best with your systematic review study and I look forward to reading the outcomes of your research.
--	--

VERSION 1 – AUTHOR RESPONSE

Reviewer 1	
The inclusion of conference abstracts and reports (not closely defined) is potentially problematic – especially with inclusion of studies in the review regardless of data being able to be extracted or of assessed study quality	The inclusion of conference papers was a ' seeking mechanism '. We wanted to access all available empirical papers on the topic which could have been missed. If conference abstracts and reports were located, it was our intention to contact the authors to request information on full data and analysis. Also transparency in methodological quality would have been made known to readers. However, this scenario did not arise.
The manuscript needs careful review of English usage and syntax. Clumsy language and grammar is frequent.	Thank you, this has been attended to: please see revised manuscript
I'm not sure why the team will be seeking ethics approval for a systematic review.	Coventry University requires every part of a PhD study (including literature reviews) to go through ethics. Please see attached pdf which confirms communication with the ethics team where the need for ethics is confirmed; titled- ethics response for BMJ open
Reviewer 2	
The PRISMA-P checklist does not contain page or line numbers to direct the reader to where in the article the PRISMA-P criteria is located, I suggest these are added	Attended to: see PRISMA-P checklist
Also the PRISMA-P checklist is presented on a Biomed Central form. I suggest you submit a standard PRISMA-P statement rather than	Attended to: see PRISMA-P checklist

one customized from a different journal	
I notice that you will be applying for ethical approvals for your study. Usually, an ethics application is not required for a systematic review as you will be dealing with secondary data. I suggest you clarify this point with your institution	Coventry University requires every part of a PhD project or any research to go through ethics. Specifically, please see pdf which confirms communication with the ethics team; titled- ethics response for BMJ open. Find attached a link to the University policy https://www.coventry.ac.uk/research/about-us/research-ethics/
You mentioned that you will be taking a mixed method systematic review approach and will analyse both qualitative and quantitative data. However, in your PRISMA-P statement under point 15a you mention that quantitative data won't be synthesized. Please could you clarify this?	15a. Describe criteria under which study data will be quantitatively synthesised. Response: No data will undergo quantitative synthesis , rather the quantitative extracted data will undergo 'qualitization synthesis' i.e. the extracted quantitative data will be converted into a qualitative data. There is no meta-analysis for this review Please see page 7 & 8 of manuscript, data transformation and data synthesis

VERSION 2 – REVIEW

REVIEWER	Naughton, Bernard Trinity College Dublin, Pharmacy and Pharmaceutical Sciences
REVIEW RETURNED	29-May-2023

GENERAL COMMENTS	Thank you for your resubmission of this article which has been improved since I last reviewed it: I still have one outstanding querie. Why did you include concept #5 ("Qualitative Research"[Mesh] OR "mixed method"[tw] OR qualitative[tw] OR quantitative[tw]. I am unsure what value this brings do you want to exclude studies that do not specifically indicate their methodology type in the title or abstract? By doing so you might exclude valuable articles. Concept #3 is also methodologically based "Survey"[tw] OR "questionnaire"[tw] OR "interview"[tw] OR "observ"[tw] OR "case stud"[tw] OR "focus group"[tw] I am not sure if you need both or either unless you have a good justification?
---

VERSION 2 – AUTHOR RESPONSE

REVIEWER'S QUERY	AUTHORS RESPONSE
Why did you include concept #5 ("Qualitative Research"[Mesh] OR "mixed method*" [tw] OR qualitative [tw] OR quantitative [tw]. I am unsure what value this brings do you want to exclude studies that do not specifically indicate their methodology type in the title or abstract? By doing so you might exclude valuable articles. Concept #3 is also methodologically based "Survey*" [tw] OR "questionnaire*" [tw] OR "interview*" [tw] OR "observ*" [tw] OR "case stud*" [tw] OR "focus group*" [tw] I am not sure if you need both or either unless you have a good justification?	We wanted to ensure maximum retrieval of empirical research studies into the topic of interest. We utilised number of approaches in the search. Searching with the MESH terms and text word (tw) optimizes search performance and attain a balance between high recall and high precision (McKibbon & Marks, 1998; Sampson et al., 2009). Tw (text word) searches the title, abstract and keyword fields so broadens the search rather than narrows it. In addition, during our search in the databases, we chose 'all text in the filter' for the concepts used hence not only restricted to title and abstract. Using methodological terms acted as a means of increasing the sensitivity, precision and specificity of the search (Bramer et al., 2018). Also, the concept #3 and #5 stems from the SPIDER framework utilised in this study. The SPIDER tool has the added advantage that it might be suitable for mixed-methods, qualitative and quantitative research search strategies, made possible by the addition of "Research type" which is concept #5 (Cooke et al., 2012). However, to account for variations in the indexing approach of different databases, individualized search strategies were developed for each source. For example, to counteract the risk of losing good or relevant articles, the search strategy in SCOPUS database was manipulated. We only did the search using 'nursing' and 'climate change' which is concept #1 and #2. Additionally, we employed forward citation search technique to locate any potentially missed relevant papers Further to the above, the database search was constructed with the support of experienced librarian. All these strategies were applied to ensure most thorough search.

- Bramer, W. M., de Jonge, G. B., Rethlefsen, M. L., Mast, F., & Kleijnen, J. (2018). A systematic approach to searching: an efficient and complete method to develop literature searches. *J Med Libr Assoc*, 106(4), 531-541. <https://doi.org/10.5195/jmla.2018.283>
- Cooke, A., Smith, D., & Booth, A. J. Q. h. r. (2012). Beyond PICO: the SPIDER tool for qualitative evidence synthesis. 22(10), 1435-1443.
- McKibbin, K. A., & Marks, S. J. E.-B. N. (1998). Searching for the best evidence. Part 1: where to look. 1(3), 68-70.
- Sampson, M., McGowan, J., Cogo, E., Grimshaw, J., Moher, D., & Lefebvre, C. (2009). An evidence-based practice guideline for the peer review of electronic search strategies. *Journal of Clinical Epidemiology*, 62(9), 944-952. <https://doi.org/https://doi.org/10.1016/j.jclinepi.2008.10.012>